# Access to primary healthcare during lockdown measures for COVID-19 in rural South Africa: an interrupted time series analysis

Mark J Siedner [1,2] John D Kraemer,[3] Mark J Meyer [4] Guy Harling [5,6]
Thobeka Mngomezulu,[7] Patrick Gabela,[7] Siphephelo Dlamini,[8] Dickman Gareta,[9]
Nomathamsanqa Majozi,[10] Nothando Ngwenya,[6] Janet Seeley [11,12]
Emily Wong,[1] Collins Iwuji,[13] Maryam Shahmanesh,[1] Willem Hanekom,[1]
Kobus Herbst[7]

For numbered affiliations see end of article.

**Correspondence to**
Dr Mark J Siedner;
mark.siedner@ahri.org

## ABSTRACT

**Objectives** We evaluated whether implementation of lockdown orders in South Africa affected ambulatory clinic visitation in rural Kwa-Zulu Natal (KZN).

**Design** Observational cohort

**Setting** Data were analysed from 11 primary healthcare clinics in northern KZN.

**Participants** A total of 46 523 individuals made 89 476 clinic visits during the observation period.

**Exposure of interest** We conducted an interrupted time series analysis to estimate changes in clinic visitation with a focus on transitions from the prelockdown to the level 5, 4 and 3 lockdown periods.

**Outcome measures** Daily clinic visitation at ambulatory clinics. In stratified analyses, we assessed visitation for the following subcategories: child health, perinatal care and family planning, HIV services, non-communicable diseases and by age and sex strata.

**Results** We found no change in total clinic visits/clinic/day at the time of implementation of the level 5 lockdown (change from 90.3 to 84.6 mean visits/clinic/day, 95% CI −16.5 to 3.1), or at the transitions to less stringent level 4 and 3 lockdown levels. We did detect a >50% reduction in child healthcare visits at the start of the level 5 lockdown from 11.9 to 4.7 visits/day (−7.1 visits/clinic/day, 95% CI −8.9 to 5.3), both for children aged <1 year and 1–5 years, with a gradual return to prelockdown within 3 months after the first lockdown measure. In contrast, we found no drop in clinic visitation in adults at the start of the level 5 lockdown, or related to HIV care (from 37.5 to 45.6, 8.0 visits/clinic/day, 95% CI 2.1 to 13.8).

**Conclusions** In rural KZN, we identified a significant, although temporary, reduction in child healthcare visitation but general resilience of adult ambulatory care provision during the first 4 months of the lockdown. Future work should explore the impacts of the circulating epidemic on primary care provision and long-term impacts of reduced child visitation on outcomes in the region.

## INTRODUCTION

COVID-19 was declared a global pandemic by WHO on 11 March 2020, and it has spared

no region of the world. Early in the epidemic, the greatest numbers of cases have been reported in Asia, Europe and North America, with more recent dissemination within Latin America and Africa.[1] Although South Africa and a handful of other low-resource settings have reported widespread epidemics, limited testing and surveillance capabilities make it difficult to assess how widely the pandemic has spread in such settings. Such regions are believed to be at particular risk of severe epidemics, due to overcrowding, lower access to clean water and sanitation services and inherent shortages in health system infrastructure for detection and management of disease.[2–9]

### Strengths and limitations of this study

► This study is strengthened by a large number of observations per clinic, collection of data using research staff and methods external to the programmatic healthcare system (which can be interrupted during external shocks), and longitudinal observation allowing comparison of the preperiod and postperiod.
► The study was conducted within the public health sector of South Africa, and is generalisable to similarly rural, resource-limited settings, and observation during the lockdown period prior to widespread local COVID-19 transmission.
► Our exposure of interest (ie, the lockdown measures) were not randomly allocated, so there is a risk, however small, that a simultaneous exogenous shock could be responsible for the changes in clinic visitation seen.
► The dataset is purely quantitative and focused on clinic visitation events, which prevents deeper exploration of the root causes of trends noted.

In response, most nations in sub-Saharan Africa have implemented non-pharmacological interventions to attempt to prevent large-scale epidemics. These measures, which include restrictions on large gatherings, work and school attendance, travel and in their most stringent forms, shelter-in-place orders, are believed to reduce disease transmission.[10–13] However, instituting these measures is also associated with deleterious economic and social, impacts, including large projected reductions in manufacturing, access to employment and basic necessities and educational advancement; and these effects appear to be greatest among those in lower-income and vulnerability categories.[14–19] Across the sub-Saharan African region, the Economic Commission for Africa projects an approximate 1.4% contraction in gross domestic product and that 25 million people are susceptible to entering extreme poverty.[20] Some have hypothesised that non-pharmaceutical interventions might be less effective in settings with large informal economies and limited ability to respond to increases in cases of severe disease,[21] and that their risks might outweigh their benefits.[22]

Of particular concern is how social fear and reduced access to basic public health services might impact morbidity and mortality for non-COVID health conditions. Modelling studies have suggested that even modest reductions in child healthcare access could result in 100 000s of additional deaths in low-income and middle-income countries.[23] Similar concerns have been raised by the Academy of Science of South Africa and others about provision of chronic disease care among adults.[24 25] UNAIDS has warned that non-pharmaceutical interventions could challenge manufacturing and supply chains of HIV therapeutics,[26] and modelling estimates suggest that such disruptions could cause as many if not more HIV-related deaths than COVID-19-related deaths.[27] Although empiric data on health outcomes remain sparse, there have been significant reductions in tuberculosis testing in South Africa during the early phases of the lockdown,[28] indicating an interruption in critical services for the most common cause of death in the country.[29] There is also historical precedent from other recent communicable disease outbreaks. Primary healthcare access was significantly impacted during prior infectious disease epidemics, such as Ebola virus disease, resulting in increases in morbidity and mortality.[30 31] Yet, whether and the extent to which similar effects will be seen during the COVID-19 epidemic is not known.

On 27 March 2020, South Africa instituted a nationwide shelter-in-place order, termed in South Africa as a level 5 lockdown. The level 5 order included closure of schools and all non-essential business, restrictions on public transport and restrictions on movement. Restrictions on movement during the level 5 lockdown specifically required that individuals remain in their place of residence, with the exceptions of 'performing an essential service, obtaining an essential good or service, collecting a social grant, pension, or seeking emergency, life-saving or chronic medical attention'. Over the following months, the restrictions gradually eased from level 5 down to 4 at the end of April and level 3 at the end of May, which corresponded with lifting restrictions on intraprovince movement, preinitiation of public transportation and allowed for reopening of schools and many business.[32 33] Because the healthcare sector was deemed an essential service throughout the entire lockdown period, no restrictions were placed on access to or delivery of healthcare services.

We sought to assess the impact of these lockdown orders in response to the COVID-19 epidemic in South Africa on access to basic healthcare services. We analysed data on clinic visitation at 11 ambulatory public health clinics in northern KwaZulu-Natal, collected routinely as part of a demographic health and surveillance system (HDSS) by the Africa Health Research Institute (AHRI). We hypothesised that there would be immediate and substantial reductions in clinic visitation after the institution of the lockdown measure, and that this would pertain to routine clinical care such as immunisations, perinatal care and chronic disease management in adults.

## METHODS
### Study setting
This analysis was conducted using data collected by the AHRI HDSS in the uMkhanyakude district of the KwaZulu-Natal Province. The HDSS comprises a complete census across a geographic area of approximately 850 km$^2$; it is a rural region with a single peri-urban centre, KwaMsane, a town of approximately 30 000 residents. The region ranks among the lowest nationwide in terms of health indicators and socioeconomic status.[34] Approximately one in five adult men and two in five adult women are living with HIV.[35] Tuberculosis incidence is among the highest in the world, and above the national average of 577 per 100 000 individuals when last measured in 2015.[36]

### Data collection
Since 2000 AHRI has collected data on births, deaths, migrations through thrice annual data collection encounters across a catchment area of 20 000 households (over 100 000 resident individuals).[34] In 2017, AHRI began placing clinic research assistants at each of the 11 government-run public health clinics in the area. These research staff operate in partnership with the Department of Health, but outside of the standard Health Management Information System (HMIS). For each person who presents to clinic, they collect demographic information and the self-reported reason(s) for the clinical visit. We link data between the HDSS and the clinic medical record system electronically using a unifying identification code for each resident of the catchment area. For this analysis, we included all individuals who presented to ambulatory clinic in the 11 regional clinics in the study catchment area during our observation period. There were no age or sex exclusions. AHRI holds memoranda of understanding with the Provincial and District Department of Health that permit extraction of health record data from

primary care and hospital sites for linkage to the household surveillance dataset.

## Study design

We conducted an interrupted time series analysis to estimate changes in clinic visitation in rural KwaZulu-Natal before and after the national lockdown implementation on 27 March 2020. To do so, we fitted linear mixed effects regression models by restricted maximum likelihood with daily clinic visits as the primary outcome of interest. Our primary exposure of interest was time period, divided into four periods: (1) the prelockdown period starting 60 days prior to the initial level 5 lockdown until 27 March 2020; (2) the level 5 lockdown period from 28 March through 30 April 2020; (3) the level 4 lockdown level from 1 May through 31 May and (4) the level 3 lockdown level from 1 June through our data abstraction date (30 June). For our primary outcome, we estimated the stepwise change in mean visits per clinic on the date of implementation lockdown level.[37] We included a fixed effect for day of the week, random clinic-specific intercepts and random clinic-specific slopes on time in our models. We excluded weekends because the study clinics do not provide non-urgent ambulatory care services on weekends. We excluded dates from observation when AHRI staff members who perform data capture for the clinic-link system were not working, including national holidays and staff trainings.

Our primary outcome of interest was the number of clinic visits for any reason per clinic. In secondary analyses, we stratified models by visit type restricted to: (1) child health visits (immunisations and growth monitoring); (2) antenatal care, postnatal care and family planning; (3) HIV services (including antiretroviral therapy initiation, antiretroviral therapy continuation and chronic care medical dispensing programme visits) and (4) chronic care of noncommunicable diseases (hypertension and diabetes). Clinic visits for more than one reason were treated as visits for both conditions. We also conducted stratified analyses by age category (<1, 1–5, 6–19, 20–45 and >45 years) and by women and men aged 15 years or older.

We performed a number of secondary and sensitivity analyses to assess model validity and robustness of our findings. In addition to estimating stepwise changes after each lockdown order and change, we estimated trends in weekly visits during each period to determine whether immediate changes were sustained over each period. We graphically depicted the residuals in the model to assess the normality assumption of our linear model structure. To check the robustness of model assumptions about changes from the prelockdown to the level 5 lockdown periods, we conducted multiple sensitivity analyses: (1) we specified a Poisson mixed effects regression model in place of a linear model; (2) we fitted linear and Poisson generalised estimating equation (GEE) models clustered by facility; (3) we specified an autoregressive covariance structure in place of an exchangeable structure in the GEE models and (4) to assess for the possibility of seasonal changes over multiple years, we constructed locally weighted scatterplot smoothing (LOWESS) plots to visually inspect clinical visitation trends over the same observation periods in 2020 vs 2018 and 2019 and fitted a difference-in-differences model that included year (2019 vs 2020) and time (characterised as prelockdown vs level 5 lockdown) to assess whether changes before and after the level 5 lockdown differed by calendar year.

Finally, we conducted an additional sensitivity analysis to assess for the possibility of in-migration into the HDSS catchment area during the lockdown period, which would potentially bias clinic visitation frequency upwards. To do so, we calculated annual visitation frequency at the 11 area clinics for each individual in the dataset for the year prior to the lockdown. We then compared the median number of annual visits per individual in the prelockdown and postlockdown periods, and the number of individuals with exactly one visit in the past year in the two periods. If significant in-migration did occur during the lockdown period, we would expect that the median number of annual visits per individual would decrease during the lockdown, whereas the number of individuals with one visit in the past 12 months would increase.

Three study investigators designed a statistical plan prior to all analyses (MJS, JDK, MJM). The initial analysis plan included fitting mixed effects models with random effects by clinic and inspecting trends in clinic visitation changes from the prelockdown to level 5 lockdown period for the overall cohort, and by visit subtype. We also initially included plans for sensitivity analyses, including fitting of generalised estimating equations and additions of random slopes to our models as robustness checks. In response to reviewer requests and with updates to the lockdown characteristics from levels 5 to 4 to 3 during the review process, we conducted a number of post hoc analyses, including construction of LOWESS plots and fitting additional models to assess for seasonal trends in visitation by year, and fitting mixed effects Poisson models. All statistical analyses were conducted using Stata and R.

## Patient and public involvement

This protocol was reviewed and approved by the AHRI Community Advisory Board, who contributed to the study design and selection of collection measures. Results of studies from the HDSS project are routinely shared with the community through public communications and road shows conducted by the AHRI Public Engagement Department. Final, all study protocols are reviewed and approved by the District and Provincial Department of Health, and AHRI holds memoranda of understanding with the Provincial and District Departments of Health that outline methods of extraction of health record data from primary care sites for linkage to the household surveillance dataset.

## RESULTS

A total of 46 523 individuals made 89 476 clinic visits between 27 January and 30 June 2020 at the 11 area clinics (table 1). Women and girls accounted for 67% (n=64 125) of visits. Approximately 9% of visits were

**Table 1** Ambulatory clinic visits at 11 region clinics in rural KwaZulu Natal during 27 January 2020–30 June 2020 by sex and age and clinic visit type

| | Total* n (%) | Male n (%) | Female n (%) | <1 year n (%) | 1–5 years n (%) | 6–19 years n (%) | 20–45 years n (%) | >45 years n (%) |
|---|---|---|---|---|---|---|---|---|
| Total visits | 89 476 (100%) | 25 318 (28.3%) | 64 125 (71.7%) | 7734 (8.6%) | 5646 (6.3%) | 7429 (8.3%) | 43 116 (48.2%) | 25 551 (28.6%) |
| Child health | 9672 (10.8%) | 4808 (49.7%) | 4846 (50.1%) | 6712 (69.4%) | 2750 (28.4%) | N/A | N/A | N/A |
| PNC and FP† | 7879 (8.8%) | 13 (0.2%) | 7866 (99.8%) | 9 (0.1%) | 2 (0.0%) | 1266 (16.1%) | 6570 (83.4%) | 32 (0.4%) |
| HIV visit‡ | 42 585 (47.6%) | 11 152 (26.2%) | 31 425 (73.8%) | 44 (0.1%) | 215 (0.5%) | 2549 (6.0%) | 27 218 (63.9%) | 12 559 (29.5%) |
| Chronic care§ | 10 052 (11.2%) | 2143 (21.3%) | 7909 (78.7%) | 1 (0.0%) | 2 (0.0%) | 11 (0.1%) | 647 (6.4%) | 9391 (93.4%) |
| Minor ailment | 2738 (3.1%) | 988 (36.1%) | 1750 (63.9%) | 133 (4.9%) | 471 (17.2%) | 384 (14.0%) | 1084 (39.6%) | 666 (24.3%) |
| All other visits | 18 043 (20.2%) | 6378 (35.4%) | 11 658 (64.6%) | 842 (4.7%) | 2222 (12.3%) | 3112 (17.3%) | 8390 (46.5%) | 3477 (19.3%) |

*Visit types are not mutually exclusive so column totals may exceed 100%.
†PNC and FP; visits for antenatal care, prenatal care and/or FP.
‡HIV visits: visits for HIV testing, antiretroviral therapy initiation, antiretroviral therapy continuation or pharmacy pick-up.
§Chronic care: visits for hypertension and/or diabetes.
FP, family planning; PNC, perinatal care.

made by individuals <1 year of age (n=4186), 1–5 years of age (n=3944) and 6–19 years of age (n=4460), respectively; whereas those aged 20–45 years accounted for 48% (n=22 231) and those over 46 years the remaining 25% of visits (n=11 702). The most common reason for a clinic visit was ART follow-up care, comprising 43% of all visits (n=38 142), followed by visits for minor ailments (18%, n=16 204), child health (9672, 11%) and hypertension (n=9273, n=10%).

There was an average of 90.3 (95% CI 67.5 to 113.2) clinic visits per day per clinic in the prelockdown period. We identified a non-significant drop in visits immediately following the start of the level 5 lockdown (−6.7 visits/ clinic/day, 95% CI −16.5 to 3.1). The small reduction seen after the level 5 period was reversed by a non-significant stepwise increase between the level 5 and level 4 periods (increase of 11.2 visits/clinic/day, 95% CI −0.5 to 23.0), and persistent clinic visitation between the end of level 4 and the start of the level 3 period (increase of 1.2 visits/ clinic/day, 95% CI −9.7 to 12.1) (table 2, figure 1). There were no significant changes in trends over time in clinic visits/week in any of the preimplementation or postimplementation periods (online supplemental table 1).

In contrast, child health visits suffered a 60% stepwise drop at the initiation of the level 5 lockdown (from 11.9 to 4.6 visits/day/clinic, mean change of −7.1 visits, 95% CI −9.0 to 5.3), but remained steady during the transition between the level 5 and level 4 (−0.5 visits/clinic/ day, 95% CI −2.7 to 1.7), and again between the level 4 and level 3 lockdowns (−0.4 visits/clinic/day, 95% CI −2.5 to 1.6) (table 2, figure 1). The reduction in child visits at the time of the level 5 lockdown occurred both among children under 1 (mean decrease of −5.4 visits, 95% CI −7.0 to 3.7) and those 1–5 years of age (mean decrease of −5.5 visits, 95% CI −6.7 to 4.4). We did detect fluctuating trends in child visitation during the lockdown periods, with increases of approximately one visit/clinic/ week in levels 5 and 3, resulting in a similar number of mean visits/clinic/day 1 month into to the level 3 lockdown on 30 June as compared with just prior to the level 5 lockdown (11.2 visits/clinic/day (95% CI 7.4 to 14.8) vs 11.9 visits/clinic/day (95% CI 8.6 to 15.1), online supplemental table 1).

In contrast to child health visits, HIV-related clinical visits for adults did not decrease between the prelockdown and level 5 lockdown period from 37.6 visits/clinic/day to 45.5 visits/clinic/day (mean change of 7.9 visits/clinic/ day, 95% CI 2.1 to 13.8), between the level 5 lockdown and level 4 lockdown period (increase of 11.1 clinic/ visits/day, 95% CI 4.1 to 18.0) or between the transition from the level 4 to level 3 (increase of 4.0 visits/clinic/ day, 95% CI −2.5 to 10.5) (table 2, figure 1). We similarly identified resilience in family planning visits over the observation period, increasing from 7.3 visits/clinic/day in the preimplementation period to 7.8 visits/clinic/day after transition to level 5 (+0.5 visits/clinic/day, 95% CI −1.0 to 2.0) to 8.9 clinic visits/day after transition to level 4 (+1.1 visits/clinic/day, 95% CI −0.7 to 3.0) and 11.0 visits/

**Table 2** Mixed effects regression model results demonstrating changes in mean visits/clinic, by visit type and demographic strata, in the prelockdown period and after each transition to level 5, level 4 and level 3 lockdowns in uMkhanyakude District, KwaZulu–Natal South Africa

| Model | Mean daily visits per clinic at time of lockdown period (intercept) | Stepwise change in visits/clinic/day at start of level 5 lockdown | P value | Stepwise change in visits/clinic/day at start of level 4 lockdown | P value | Stepwise change in clinic visits/clinic/day at start of level 3 lockdown | P value |
|---|---|---|---|---|---|---|---|
| Total visits | 90.3 (67.1 to 113.5) | −6.7 (−16.4 to 3.0) | 0.18 | 11.3 (−0.3 to 22.9) | 0.06 | 1.2 (−9.6 to 12.0) | 0.83 |
| Child health* | 11.9 (8.6 to 15.1) | −7.1 (−8.9 to −5.3) | <0.001 | −0.5 (−2.6 to 1.6) | 0.65 | −0.4 (−2.4 to 1.6) | 0.67 |
| PNC and FP† | 7.3 (4.2 to 10.3) | 0.5 (−1.0 to 2.0) | 0.51 | 1.1 (−0.7 to 2.9) | 0.22 | 2.0 (0.3 to 3.7) | 0.02 |
| HIV visits‡ | 37.5 (24.4 to 50.7) | 8.0 (2.3 to 13.7) | 0.01 | 11.0 (4.2 to 17.8) | 0.001 | 4.0 (−2.3 to 10.3) | 0.22 |
| Chronic care§ | 9.5 (7.1 to 11.8) | −0.3 (−1.9 to 1.3) | 0.70 | −0.15 (−2.1 to 1.8) | 0.88 | 0.4 (−1.4 to 2.2) | 0.65 |
| Men ≥15 | 15.4 (10.8 to 19.9) | 1.5 (−0.7 to 3.6) | 0.17 | 2.3 (−0.2 to 4.9) | 0.07 | 0.8 (−1.5 to 3.2) | 0.50 |
| Women ≥15 | 52.9 (38.1 to 67.7) | 3.2 (−3.3 to 9.6) | 0.33 | 9.6 (1.9 to 17.3) | 0.01 | 2.1 (−5.1 to 9.3) | 0.57 |
| Age <1 | 10.6 (7.6 to 13.5) | −5.3 (−6.9 to −3.7) | <0.001 | 0.2 (−1.6 to 2.1) | 0.83 | −0.4 (−2.1 to 1.4) | 0.67 |
| Age 1–5 | 8.9 (7.4 to 10.5) | −5.6 (−6.7 to −4.4) | <0.001 | −0.2 (−1.5 to 1.1) | 0.80 | −0.8 (−2.1 to 0.4) | 0.18 |
| Age 6–19 | 8.0 (6.2 to 9.8) | −0.7 (−1.9 to 0.6) | 0.29 | 1.5 (0.0 to 3.0) | 0.05 | −0.1 (−1.5 to 1.3) | 0.91 |
| Age 20–45 | 39.3 (26.3 to 52.2) | 4.4 (−0.7 to 9.5) | 0.09 | 7.5 (1.4 to 13.6) | 0.02 | 2.2 (−3.5 to 7.8) | 0.45 |
| Age >45 | 25.2 (19.2 to 31.3) | 0.4 (−3.0 to 3.7) | 0.83 | 2.9 (−1.1 to 6.9) | 0.15 | 0.5 (−3.2 to 4.3) | 0.78 |

*Child health: visits for immunisations and growth monitoring.
†PNC and FP: visits for, antenatal care, prenatal care and/or FP.
‡HIV visits: visits for HIV testing, antiretroviral therapy initiation, antiretroviral therapy continuation or pharmacy pick-up.
§Chronic care: clinical visits for hypertension and/or diabetes.
FP, family planning; PNC, perinatal care.

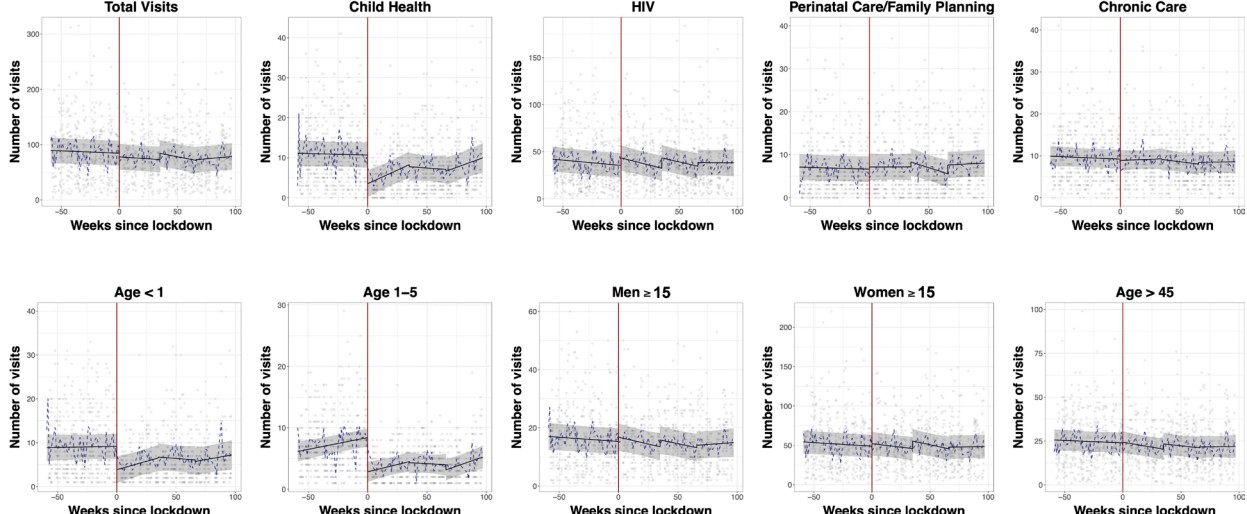

**Figure 1** Ambulatory clinic visitation before and after the nationwide lockdown in South Africa at 11 outpatient clinics in rural uMkhanyakude District, KwaZulu-Natal South Africa. Scatter plots represent mean clinic visitation at each clinic on weekdays during the observation period. The black fit line represents the mean visitation across all clinics estimated by postregression margins from a linear regression model, with a regression discontinuity coefficient at the date of the lockdown (27 March 2020, red line). Grey bars represent 95% CIs. The dotted blue line represents the geometric mean of the number of visits across all clinics on each day.

clinic/day after transition to level 3 (+2.0 visits/clinic/day, 95% CI 0.3 to 3.7) for a 66% total increase from the preperiod. We did not detect changes in clinic visitation for chronic non-communicable diseases, or more broadly in clinic visitation by men or women 15 years or older.

Graphical depictions of residuals from our linear models demonstrated normally distributed residuals for the total visit, HIV and child visit models, supporting the specification of a linear model to estimate trends in clinic visitation (online supplemental figure 1). We found no evidence of changes in clinic visitation during this same period in 2019 to suggest seasonal effects, either graphically in LOWESS plots, or in difference-in-differences in models including both 2019 and 2020 with prelockdown and postlockdown periods (table 3, online supplemental figure 2). Results were robust to modelling assumptions in the sensitivity analyses (table 3).

Finally, we did not detect evidence of meaningful in-migration during the lockdown period. The median number of visits in the past year per individual attending the clinic slightly increased from the prelockdown period to the lockdown period (mean 5.8 (SD 0.02) vs 5.9 (SD 0.03), p<0.001). This pattern was similar among people attending clinic for HIV-specific visits (mean 6.5 (SD 0.02) vs mean 6.6 (SD 0.04), p=0.01). The number of people with exactly one visit in the past year also did not meaningfully increase during the observation period with 1960 (February), 2115 (March), 1573 (April), 1893 (May) and 1986 (June) visits made by individuals with exactly one annual clinic visit over the prior 12 months.

## DISCUSSION

We found evidence of a significant drop in visits for childcare alongside sustained visitation in HIV and adult ambulatory clinic utilisation in a rural area of South Africa during the national lockdown for the COVID-19 epidemic. Notably, visits for chronic disease, such as hypertension and diabetes, perinatal care and family planning remained reasonably constant or modestly increased. However, child health visits for immunisations and growth monitoring dropped immediately by over 50% after the start of the lockdown. With gradual increases over time during level 5 and level 3 lockdown, child visits largely returned to prelockdown levels in June, approximately 3 months after the lockdown began. We noted an estimated 20% increase in clinic visits for HIV immediately after the lockdown and suspect this might have reflected an urgency to collect medications prior to an anticipated interruption in clinic access or medication availability and/or national programmatic efforts to accelerate transitions to a new first-line regimen.[38] These results demonstrate concerning trends about reductions in preventative child care during the lockdown period. However, they also appeared to disprove our hypothesis about clinic visitation in adults, and potentially demonstrate a resilience in the healthcare sector during a period of concern for access to chronic and essential basic health services.

The key demographic population in our study that experienced significant drops in clinic visitation was children. Child health visits appeared to have modestly rebounded during the lockdown, and eventually return to their prelockdown state. Although these data do not suggest the cause of reduced visitation in children, multiple possible factors might be considered. We hypothesise that limited options for childcare for families with multiple children might prevent caregivers from being able to bring individual children to clinic. Moreover, in

**Table 3** Sensitivity analyses, demonstrating results of the main regression model and alternate models

| All visits | Mean daily visits per clinic at time of lockdown period | Stepwise change in clinic visits/day at start of level 5 lockdown | P value |
|---|---|---|---|
| Primary model | 90.3 (67.1 to 113.5) | −6.7 (−16.4 to 3.0) | 0.18 |
| Poisson* mixed effects model | 92.0 (59.3 to 124.7) | −6.9 (−11.0 to −2.8) | 0.001 |
| Linear GEE (exchangeable correlation matrix) | 89.2 (67.0 to 111.4) | −6.4 (−16.8 to 4.08) | 0.23 |
| Poisson GEE* (exchangeable correlation matrix) | 89.2 (84.7 to 93.6) | −6.6 (−8.7 to −4.5) | <0.001 |
| Linear GEE (autoregressive correlation matrix) | 90.2 (73.2 to 107.2) | −5.4 (−27.4 to 16.6) | 0.63 |
| Poisson GEE* (autoregressive correlation matrix) | 88.4 (85.0 to 91.9) | −5.0 (−9.4 to −0.6) | 0.03 |
| Difference-in-differences† | 96.3 (63.6 to 129.0) | 3.4 (−5.5 to 12.4) | 0.45 |
| Childcare visits | | | |
| Primary model | 11.9 (8.6 to 15.1) | −7.1 (−8.9 to −5.3) | <0.001 |
| Poisson* mixed effects model | 12.3 (7.1 to 17.5) | −7.7 (−11.1 to −4.4) | <0.001 |
| Linear GEE (exchangeable correlation matrix) | 11.8 (8.6 to 15.0) | −7.1 (−9.0 to −5.2) | <0.001 |
| Poisson GEE* (exchangeable correlation matrix) | 12.0 (10.5 to 13.5) | −7.5 (−8.4 to −6.6) | <0.001 |
| Linear GEE (autoregressive correlation matrix) | 11.9 (9.5 to 14.4) | −6.4 (−9.9 to −2.9) | <0.001 |
| Poisson GEE* (autoregressive correlation matrix) | 11.9 (10.7 to 13.1) | −6.7 (−8.1 to −5.4) | <0.001 |
| Difference-in-differences† | 11.8 (8.0 to 15.7) | −4.0 (−5.5 to −2.5) | <0.001 |
| HIV visits | | | |
| Primary model | 37.5 (24.4 to 50.7) | 8.0 (2.3 to 13.7) | 0.01 |
| Poisson* mixed effects model | 39.2 (22.7 to 55.8) | 9.0 (4.5 to 13.5) | <0.001 |
| Linear GEE (exchangeable correlation matrix) | 37.7 (25.3 to 50.1) | 8.1 (2.2 to 14.0) | 0.007 |
| Poisson GEE* (exchangeable correlation matrix) | 37.7 (34.8 to 40.6) | 8.7 (7.2 to 10.3) | <0.001 |
| Linear GEE (autoregressive correlation matrix) | 38.9 (29.3 to 48.5) | 6.1 (−6.2 to 18.5) | 0.33 |
| Poisson GEE* (autoregressive correlation matrix) | 37.9 (35.7 to 40.1) | 5.7 (2.6 to 8.8) | 0.002 |
| Difference-in-differences† | 43.6 (25.1 to 62.1) | 4.8 (−0.5 to 10.1) | 0.08 |

*Poisson GEE results are presented as predictive margins and marginal effects so they represent changes on the same additive scale as the linear models.
†Difference-in-differences estimates are estimated as the mean of the level 5 lockdown period minus the mean of the prelockdown period, comparing 2020 with 2019. Estimates are based on a period-by-year interaction term fit via linear mixed models.
.GEE, generalised estimating equations.

contrast to, for example, HIV wellcare visits, well child visits rarely involve medication refills so might be prioritised lower for families. Whatever the cause, our findings are in keeping with data from elsewhere. In Hangzhou, China, paediatric healthcare visits dropped by nearly 75% during the peak of the epidemic and lockdown periods.[39] In the USA, vaccination rates in children substantially declined after a national emergency was declared in response to the COVID-19 epidemic.[40] Modelling analyses using Lived Saves Tool have suggested that a 15% reduction in maternal and child health coverage could result in over 250 000 additional deaths.[23] WHO has also projected significant increases in deaths due to malaria in children under 5 in endemic regions with disruptions in malaria care and insecticide-treated bednet distribution.[41] Although empiric data on healthcare access in South Africa remain scarce, work to date has suggested significant reductions in tuberculosis testing in laboratory databases, and reports of interruptions in care in

community-based surveys.[28 42] Previous disease epidemics in sub-Saharan Africa have also been associated with lapses in primary care access, and drops in facility based births and child healthcare access.[30 31 43 44] Consequently, future work should investigate the impacts of even modest drops in vaccination rates and child health outcomes, to better assess whether the drop we identified resulted in long-term health effects, and whether catch-up vaccination campaigns might help limit the fallout of such interruptions.[45]

Maintaining healthcare access during the epidemic requires a careful balance of primary healthcare provision and protection of vulnerable populations from COVID-19 infection. In other settings, there have been multiple reports of late and severe presentations to care for non-COVID-19 conditions, putatively due to decreased access to care or fear of nosocomial infection at healthcare facilities.[46–48] At the time of our data abstraction at the end of June during the level 3 lockdown period, fewer than

200 cases of COVID-19 infection had been reported in uMkhanyakude District.[49] Thus, our data largely reflect impacts of lockdown measures prior to an epidemic with significant local transmission. Clinics in this district instituted symptom screening at the entryway to clinics, with referral of individuals meeting criteria for persons under investigation to regional COVID-19 testing centres. Future work should revisit the impacts of the epidemic itself on access to primary healthcare.

The COVID-19 epidemic has also led to calls for decentralised care to minimise exposure for high-risk populations,[50] including those with chronic non-communicable disease, HIV, a history of tuberculosis-related lung disease and those of older ages. The lockdown was instituted rapidly in South Africa, before substantial decentralised care systems could be put in place. However, an important unanswered question is how such programmes will affect access to care and epidemic transmission in high-risk populations, including the elderly and those with immunosuppressing conditions.

Our study should be interpreted within the context of the relatively short period (3 months) of the lockdown in South Africa. As a result, we are not yet able to assess long-term repercussions from disruptions to income or from the epidemic itself, or long-term effects of lapses in primary care and vaccination on health outcomes, and our results should not be generalised over longer time horizons. It is expected that economic barriers to healthcare utilisation will increase as the epidemic's effects persist over time, including secondary effects from non-pharmaceutical interventions. These effects are likely to fall most heavily on those in the informal economy.[51] South Africa has taken steps to increase social support to counteract economic disruption from the epidemic and control measures.[52] Mitigating long-term consequences may require governments and development partners to increase access to employment and other social support services during the epidemic.

Our study had multiple strengths. First, our data collection procedures are led by research staff who remained in place during the lockdown period, so these data are not affected by barriers to data collection (eg, interruptions in staff transportation or workplace access). This is important, since many routine health information systems could be expected to suffer lapses during external shocks to the healthcare system. Second, our study was able to access data collected across 11 clinical centres within a large HDSS, which provided significant power to detect even small interruptions to healthcare access. A key potential limitation to our study is that it is predicated on the assumption that there were no other external factors that would have caused interruptions to the healthcare system on or after 27 March 2020 (eg, power outage, inclement weather). We are unaware of any such shock and believe this to be a minor risk. We saw no evidence of a seasonal effect after comparing our results with similar time periods in 2019. Our analysis should also be interpreted within the context of our study area—one with approximately 200 reported cases of COVID-19 at the time of the analysis, but in a nation with a large epidemic (approximately 200 000 cases as of early July) with established local transmission in other areas.

In summary, we report a reduction in child wellcare visitation but resilience of the adult ambulatory healthcare system during the early COVID-19 epidemic and lockdown period in rural South Africa. Future work should establish if these trends are maintained, and particularly monitor access to childcare and immunisations as a result of the trends reported in this study. Finally, in rural South Africa and similar areas, efforts to balance ongoing provision of essential preventative and chronic healthcare services might be needed to ensure healthcare access remains intact while preventing nosocomial spread of COVID-19 among high-risk populations.

**Author affiliations**
[1]Clinical Research Department, Africa Health Research Institute, Durban, Kwa-Zulu Natal, South Africa
[2]Department of Medicine, Harvard Medical School, Boston, Massachusetts, USA
[3]Department of Health Systems Administration, Georgetown University, Washington, District of Columbia, USA
[4]Department of Mathematics and Statistics, Georgetown University, Washington, District of Columbia, USA
[5]Institute for Global Health, University College London, London, UK
[6]Department of Social Sciences, Africa Health Research Institute, Durban, Kwa-Zulu Natal, South Africa
[7]Department of Population Research, Africa Health Research Institute, Durban, Kwa-Zulu Natal, South Africa
[8]Department of Nursing, Africa Health Research Institute, Durban, Kwa-Zulu Natal, South Africa
[9]Research Data Management, Africa Health Research Institute, Durban, Kwa-Zulu Natal, South Africa
[10]Public Engagement, Africa Health Research Institute, Durban, Kwa-Zulu Natal, South Africa
[11]Research Unit on AIDS, Medical Research Council and Ugandan Virus Research Institute, Entebbe, Uganda
[12]Department of Global Health and Development, London School of Hygiene and Tropical Medicine, London, UK
[13]Department of Sexual Health and HIV Medicine, Brighton and Sussex Medical School, Brighton, UK

**Acknowledgements** The authors would like to thank the outstanding field staff at the Africa Health Research Institute for their work in conducting the demographic health survey field work, as well as the community and the study participants for their time and efforts. The population HIV surveillance is a node of the South African Population Research Infrastructure Network (SAPRIN) and funded by the South African Department of Science and Innovation and hosted by the South African Medical Research Council. The authors would also like to thank to Zahra Reynolds for her assistance in organising elements of the manuscript for submission.

**Contributors** KH, MJS, MJM, JS, DG, GH, EW, JDK, CI, MS, NN, TM, PG, SD, WH, NM: conceptualisation ideas; formulation or evolution of overarching research goals and aims. KH, DG, MJS: data curation management activities to annotate, maintain research data for initial use and later reuse. MJS, JDK, MJM, GH: formal analysis application of statistical, mathematical, computational or other formal techniques to analyse or synthesise study data. KH, WH: funding acquisition of the financial support for the project leading to this publication. KH, MJS, CI, MS: investigation conducting a research and investigation process, specifically performing the experiments, or data/evidence collection. MJS, MJM, JDK: methodology development or design of methodology; creation of models. KH, JS, MS, NN, NM: project administration management and coordination responsibility for the research activity planning and execution. KH, JS, MS, NN, NM: resources provision of study materials, reagents, materials, patients, laboratory samples, animals, instrumentation, computing resources or other analysis tools. KH:

software programming, software development; designing computer programs; implementation of the computer code and supporting algorithms; testing of existing code components. KH, WH: supervision oversight and leadership responsibility for the research activity planning and execution, including mentorship external to the core team. KH, MJS, MJM, JDK: validation verification, whether as a part of the activity or separate, of the overall replication/reproducibility of results/experiments and other research outputs. MJS, MJM, JDK: visualisation preparation, creation and/or presentation of the published work, specifically visualisation/data presentation. MJS: writing—original draft preparation creation and/or presentation of the published work, specifically writing the initial draft.KH, MJS, MJM, JS, DG, GH, EW, JDK, CI, MS, NN, TM, PG, SD, WH, NM: writing—review and editing preparation, creation and/or presentation of the published work by those from the original research group, specifically critical review, commentary or revision—including prepublication or postpublication stages.

**Funding**  Wellcome Trust. 082384/Z/07/Z/. Willem Hanekom Wellcome Trust and Royal Society. 210479/Z/18/Z. Guy Harling Shahmanesh is supported by National Institutes of Health under award number 5R01MH114560-03, Bill & Melinda Gates Foundation, Grant Number OPP1136774 and OPP1171600.National Human Genome Research Institute of the National Institutes of Health (NIH) under Award Number U24HG006941, South African Medical Research Council (MRC-RFA-UFSP-01-2013/UKZN HIVEPI) and the South African Department of Science and Innovation (DSI). MJS is supported by the National Institutes of Health (R01 AI124718, R01 AG059504).

**Competing interests**  None declared.

**Patient and public involvement**  Patients and/or the public were involved in the design, or conduct, or reporting, or dissemination plans of this research. Refer to the 'Methods' section for further details.

**Patient consent for publication**  Not required.

**Ethics approval**  The protocol was reviewed and approved by the University of KwaZulu-Natal Biomedical Research Ethics Committee under reference BE290/16 and the KwaZulu Department of Health Research Committee.

**Provenance and peer review**  Not commissioned; externally peer reviewed.

**Data availability statement**  Data are available on reasonable request. Data from the AHRI HDSS are publicly available on request to the AHRI research repository which can be made here: https://data.ahri.org/index.php/auth/login/?destination.

**ORCID iDs**
Mark J Siedner http://orcid.org/0000-0003-3506-842X
Mark J Meyer http://orcid.org/0000-0003-3942-9675
Guy Harling http://orcid.org/0000-0001-6604-491X
Janet Seeley http://orcid.org/0000-0002-0583-5272

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
