## [Reviewer comments · BMJ Open]

ARTICLE DETAILS

TITLE (PROVISIONAL)	Access to primary healthcare during lockdown measures for COVID-19 in rural South Africa: an interrupted time series analysis
AUTHORS	Siedner, Mark J.; Kraemer, John D.; Meyer, Mark J.; Harling, Guy; Mngomezulu, Thobeka; Gabela, Patrick; Dlamini, Siphephelo; Gareta, Dickman; Majozi, Nomathamsanqa; Ngwenya, Nothando; Seeley, Janet; Wong, Emily; Iwuji, Collins; Shahmanesh, Maryam; Hanekom, Willem; Herbst, Kobus

VERSION 1 - REVIEW

REVIEWER	BRADLEY H WAGENAAR University of Washington United States
REVIEW RETURNED	15-Jun-2020

GENERAL COMMENTS	Overall: The authors conducted a study using routine data to assess the impact of a COVID-19 lockdown on clinic visits in rural South Africa. The analysis is relevant and timely considering the emerging data on the COVID-19 pandemic. The article is well written and the interrupted time series design is well suited for this kind of analysis. In general, the methods are strong but addressing several considerations could make the methods more robust and may change the results. Several clarifications and methodological concerns are outlined below: Abstract Design section. This is not a longitudinal cohort study, nor is it prospective. A prospective cohort study normally means recruiting individuals exposed and unexposed to a given risk factor and then followed forward in time prospectively to examine whether disease is present or absent. This is a retrospective quasi-experimental or non-randomized design. Some may call this a "natural experiment". Please update the design to either call this a non-randomized experiment, a quasi-experimental design, or a "natural experiment". Missing a period at the end of the sentence. Setting. Missing a period at the end of the sentence. Exposure. Given the availability of continuous data over time, why only use 60 days pre-intervention data? Missing period at end of sentence. Results. Unclear description of findings. What does: "we found a
--

significant increase in HIV visits at the lockdown” mean? Is this an immediate level change in the trend of the series? A change in trend? Is a reduction of 7.2 visits a day in child healthcare visits practically significant? What % reduction does this represent?

Conclusions. Why do the authors discuss decentralizing chronic care when this was not studied in the present study? I would focus on how the results inform pandemic response and ongoing response to the COVID-19 pandemic.

Introduction

Lines 16-20. Would be good to more explicitly define what impacts these non-pharmacological interventions might have, especially in terms of their risks outweighing their benefits. I assume the authors refer to phenomena such as lost wages, mental health implications of social isolation, and decreases in healthcare visits, but this should be stated. Why might these be less effective in these settings? The authors go on to explain the health implications in the next paragraph but the rest is unclear.

Line 36. Important to define “restrictions on movement”. Need to fully define what was the nature of this lockdown. Stay-at-home? Essential trips only?

Methods

Line 75. A number of concerns here with the overall study design:

1. Why do the authors only use data for 60-days prior to the lockdown date? Without a full year of data it will be impossible for the authors to eliminate seasonal trends which are very strong in these similar analyses. It seems like many more months of data would be available. See:

<https://journals.plos.org/plosmedicine/article?id=10.1371/journal.pmed.1002508>.

2. Why did the authors use a linear regression model to examine these count data? One would expect a Poisson or Negative Binomial model that would allow the variance to be proportional to the mean in some way. In addition, this would allow the examination of rates per population covered. Linear models do not have an intuitive residual variance estimation, can get negative predictions or counts, and do not account for difference in underlying population distributions across your 11 clinics. Please advise.

3. I do not support excluding weekends just because “most visits do not occur on weekends”. Why not just include them?

4. Including fixed effects by day do not seem the most important to consider. What would be most important would be to eliminate seasonality at the monthly level, not weekly variations. With only 60 days of data pre-COVID, the authors are fully unable to account for seasonal variations which may be fully responsible for the changes they are seeing. This is a major limitation of these analyses.

In the abstract as well as line 84. This is not a regression discontinuity design. In regression discontinuity, data would be assigned to intervention or control based on a cut-off score, and then analyzed in a way that compares data points just above and below the cut-off. THIS design depends on the exchangeability of the data close to the cutoff point to draw inference, where the idea in the interrupted time series design is to draw inference from all

available data. So this study needs to be characterized as an uncontrolled interrupted time series analysis, not a regression discontinuity. Furthermore, this citation that is used for justification doesn't support the claim. In lines 84-85, it is the time-series design itself that allow you to investigate the immediate level change as well as the trend change.

Lines 102-108. Testing for autocorrelation by adding random slopes for time does not make sense. This allows for each day of the week to have a different time trend, but doesn't account for residual autocorrelation in the error term. Autocorrelation and differential trends by clinic are separate statistical and theoretical issues. I am confused. A suggestion to examine autocorrelation would be to specify an AR1 error term in the model. The authors could also add robustness to the discussion of autocorrelation by examining a PACF plot of the final model, which would reveal the presence of any additional residual autocorrelation that isn't being handled by the model.

Also, the authors fit GEE and poisson GEE models for sensitivity analysis. What type of working correlation matrix was used in the GEE models? Exchangeable? AR1? Fitting a GEE model is quite a different approach and results in a marginal model, whereas the GLMMs are conditional models. A more rational sensitivity analysis would be to fit Poisson or Negative Binomial GLMMs and compare them to the Linear GLMM model. Please advise.

Lines 102-8. Sensitivity analysis that aims to allow for non-linear functional form by adding a quadratic term to the model: This would allow curvature in the data, but I'm not convinced this is the most appropriate way to assess for non-linearity, especially given the scatterplots don't display a non-linear relationship. Furthermore, if the relationship pre-COVID-19 is a non-linear relationship, the ability to forecast this as a counterfactual for "what would have happened" in the absence of COVID-19 is compromised.

General. Please provide your full statistical model to make clear how you parameterized the time-series elements of your effect (immediate level change and slope change).

General. Were all of these analyses post-hoc or pre-specified? Please make absolutely clear what was pre-specified and what was decided upon after graphing the data, looking at the data, fitting alternative models, etc. A roadmap of exactly the steps the authors took to the analysis would be helpful.

Table 2. Please modify the way you describe your immediate level change. The way it is currently described "change in mean clinic visits per day" makes it sound like a change in slope rather than an immediate change in level. It is just the mean change in clinic visits immediately after the lockdown (change in intercept). There is no time component.

Supplementary Table 3. It appears as though the Poisson GEE model is highly significant, where the primary model, random slope model, and Linear GEE models are not significant. Thus, it seems possible that many of the changes you declare are "not statistically significant" may in fact have been significant had you used a count-based model rather than a linear model. Please advise. One issue with Poisson models is overdispersion. I would highly suggest using

	either a quasi-Poisson model that allows for overdispersion, or a negative binomial model that builds the flexibility for overdispersion in the model. Figures on Page 32. I assume the line represents the overall mean of the model and the shaded gray represents Cis? Please improve these models by doing the following:  1. Add the true predicted value from your model which would include the daily fixed effects (this should vary by day). 2. Keep the overall linear mean (so the daily fixed effect should vary around the overall mean) 3. Add in the raw mean value by day across all clinics, rather than a forest plot of all observations. These three changes will allow us to see whether your Cis are appropriate and whether your model is adequately fitting to the raw mean over time. The current figures do not allow this (this is essentially the best "validation" one can do in a time-series analysis. Discussion line 192. "expected 20% increase in clinic visits for HIV". Please replace with "estimated 20% increase in clinic visits for HIV"? Or observed? It is not expected, which would refer to a value in the future or an out of sample prediction. Discussion generally. The authors should add a discussion as to why they believe clinic visits decreased so dramatically for ages ≤ 5 but remained constant for other ages?
--	---

REVIEWER	Sunday Olawale Onagbiye University of the Western Cape, South Africa
REVIEW RETURNED	21-Jun-2020

GENERAL COMMENTS	In line 27: this statement seems standalone and seems not related to the outcome investigated and should be removed for clarity. on the other hand since the the study is not focusing on Ebola. In line 33: The author seems to mostly attached social distancing to level 5 lock-down alone. it is important for the author to understand that social distancing is still existing at level 3 lock-down, although a bit of relaxing the lockdown measures. This statement needs to be rephrased for clarity for those who may the reader of already a good manuscript. In line 36: This statement seems not so important here. This is what was already known as essential services. In line 38: Before the start of the statement "We sought.....before the next statement that leads to why the study is conducted, I suggest the author should add 1-2 paragraph on access to primary health care in south Africa before lockdown, challenges during the lock-down, and this can then be followed with what was done in the study. This could later inform the re-organisation of the introduction section. In line 51: the "p" should be capitalized In line 66: this statement should be rephrased In line 123: contributed "to" the study design and collection measures (delete input on) in line 142: "years old" should come after (n=3173). in line 143: the writing of result section should follow the same
--

	pattern for clarity. it is good to write the percentage accounted for follow by number of participants e.g. 47% (n=17,226). This should be considered for others. Same is applicable for the regressions results that follows. in line 231: this statement should be referenced or the source should be provided. in line 234: this statement is not clear and should be given attention in line 248: instead of the word "will", I suggest the author use the word "may" in line 252: Not sure if this could be a strong motivation for strength of the study. the author should think about this. in line 271: "here" should be written as "in this study" in line 272: the final part as highlighted here does not seems to be a suggestion. I am also concerned about the word "will" instead of using words such as may; might; or could. This should be rephrased. In table 1: the author should check for the correctness and accuracy of the percentages, especially when it is added up which must be 100%. e.g. in a situation where the add up could be 100.2% instead of 100%. Lastly, this article should be language edited by English language editor or professional in the field. Importantly, the professional in the field would be preferred in order not to loose the scientific writing and language.
--	--

VERSION 1 – AUTHOR RESPONSE

Reviewers' Comments to Author:

Reviewer: 1

Reviewer Name: BRADLEY H WAGENAAR

Institution and Country: University of Washington, United States

Please state any competing interests or state 'None declared': None

Overall: The authors conducted a study using routine data to assess the impact of a COVID-19 lockdown on clinic visits in rural South Africa. The analysis is relevant and timely considering the emerging data on the COVID-19 pandemic. The article is well written and the interrupted time series design is well suited for this kind of analysis. In general, the methods are strong but addressing several considerations could make the methods more robust and may change the results. Several clarifications and methodological concerns are outlined below:

Reviewer Comment

Abstract Design section. This is not a longitudinal cohort study, nor is it prospective. A prospective cohort study normally means recruiting individuals exposed and unexposed to a given risk factor and then followed forward in time prospectively to examine whether disease is present or absent. This is a retrospective quasiexperimental or non-randomized design. Some may call this a "natural experiment". Please update the design to either call this a non-randomized experiment, a quasi-experimental design, or a "natural experiment".

Response:

We thank the reviewer for this comment. We have updated the title according to their suggestion. All individuals in this analysis are enrolled in a longitudinal, population-based demographic health and surveillance program with three survey encounters per year and clinic records linked to their demographic dataset. We agree with the reviewer that a quasi-experimental design is a more accurate description of the analysis we undertook and have renamed the manuscript as:

Access to primary healthcare during lockdown measures for COVID-19 in rural South Africa: an interrupted time series analysis

Reviewer Comment

Setting. Missing a period at the end of the sentence.

Response:

We have added this period.

Reviewer Comment

Exposure. Given the availability of continuous data over time, why only use 60 days pre-intervention data? Missing period at end of sentence.

Response:

We have restricted the analysis to the 60-days prior to the lockdown because we do not collect clinic data in for the lion share of the months of December or January, when research staff are away for the holiday season and/or clinics are closed. We have however added prior years to the analysis to assess for seasonality as described in greater detail below.

Reviewer Comment

Results. Unclear description of findings. What does: "we found a significant increase in HIV visits at the lockdown" mean? Is this an immediate level change in the trend of the series? A change in trend?

Response:

We thank the review for this and have improved the clarity of our results to reflect how changes measured were at the time of implementation of the level 5 lockdown. We have also added additional data to assess changes to the level 4 and 3 lockdowns which have occurred since our initial submission:

We found no change in total clinic visits/clinic/day at the time of implementation of the level 5 lockdown (change from 90.3 to 84.6 mean visits/clinic/day, 95%CI -16.5, 3.1), or at the transitions to less stringent level 4 and 3 lockdown levels. We did detect a greater than 50% reduction in child healthcare visits at the start of the level 5 lockdown from 11.9 to 4.7 visits/day (-7.1 visits/clinic/day, 95%CI -8.9, -5.3), both children <1 and children 1-5, but an eventual return to pre-lockdown levels three months later. In contrast, we found no drop in clinic visitation in adults at the start of the level 5 lockdown, or related to HIV care (from 37.5 to 45.6, 8.0 visits/clinic/day, 95%CI 2.1, 13.8).

Reviewer Comment

Is a reduction of 7.2 visits a day in child healthcare visits practically significant? What % reduction does this represent?

Response:

We have added context to this statement:

We did detect a greater than 50% reduction in child healthcare visits at the start of the level 5 lockdown from 11.9 to 4.7 visits/day (-7.1 visits/clinic/day, 95%CI -8.9, -5.3), both for children

Reviewer Comment

Conclusions. Why do the authors discuss decentralizing chronic care when this was not studied in the present study? I would focus on how the results inform pandemic response and ongoing response to the COVID-19 pandemic.

Response:

We have removed the comment about decentralization and focused more on the findings from this work:

Future work should explore the impacts of the circulating epidemic on primary care provision and longerterm impacts of reduced child visitation on outcomes in the region.

Reviewer Comment

Introduction

Lines 16-20. Would be good to more explicitly define what impacts these non-pharmacological interventions might have, especially in terms of their risks outweighing their benefits. I assume the authors refer to phenomena such as lost wages, mental health implications of social isolation, and decreases in healthcare visits, but this should be stated. Why might these be less effective in these settings? The authors go on to explain the health implications in the next paragraph but the rest is unclear.

Response:

We have added context to the introduction including a more thorough review of data on the economic and social impacts of physical distancing measures:

However, instituting these measures is also associated with deleterious economic and social, impacts, including large projected reductions in manufacturing, access to employment and basic necessities, and educational advancement; and these effects appear to be greatest among those in lower income and vulnerability categories.¹⁴⁻¹⁹ Across the sub-Saharan African region, the Economic Commission for Africa projects an approximate 1.4% contraction in gross domestic product and that 25 million people are susceptible to entering extreme poverty.²⁰

With additional data since our first submission, we have also expanded upon the impacts on the South African health system:

Of particular concern is how social fear and reduced access to basic public health services might impact morbidity and mortality for non-COVID health conditions. Modeling studies have suggested that even modest reductions in child healthcare access could result in 100,000s of additional deaths in low and middle-income countries.²³ Similar concerns have been raised by the Academy of Science of South Africa and others about provision of chronic disease care among adults.^{24 25} UNAIDS has warned that nonpharmaceutical interventions could challenge manufacturing and supply chains of HIV therapeutics,²⁶ and modeling estimates suggest that such disruptions could result in as many or more HIV-related deaths than COVID-19-related deaths.²⁷ Although empiric data on health outcomes remain sparse, there have been significant reductions in tuberculosis testing in South Africa during the early phases of the lockdown,²⁸ indicating an interruption in critical services for the most common cause of death in the country.²⁹ Primary healthcare access was significantly impacted during prior infectious disease epidemics, such as Ebola virus disease, resulting in increases in morbidity and mortality.^{30 31} Yet, whether and the extent to which similar effects will be seen during the COVID-19 epidemic is not known.

Reviewer Comment

Line 36. Important to define “restrictions on movement”. Need to fully define what was the nature of this lockdown. Stay-at-home? Essential trips only?

Response:

We have further clarified the restriction on movement order in level 5 and added context about the level 3 and 4 orders, which we have added to this updated manuscript, along with references which give greater detail on each:

The level 5 order included closure of schools and all non-essential business, restrictions on public transport, and restrictions on movement. Restrictions on movement during the level 5 lockdown specifically required that individuals remain in their place of residence, with the exceptions of “performing an essential service, obtaining an essential good or service, collecting a social grant, pension, or seeking emergency, life-saving or chronic medical attention.” Over the following months, the restrictions gradually eased from level 5 down to 4 at the end of April and level 3 at the end of May, which corresponded with lifting restrictions on intra-province movement, preinitiation of public transportation, and allowed for reopening of schools and many business. 32, 33 Because the healthcare sector was deemed an essential service throughout the entire lockdown period, no restrictions were placed on access to or delivery of healthcare services.

Reviewer Comment

Methods

Line 75. A number of concerns here with the overall study design:

1. Why do the authors only use data for 60-days prior to the lockdown date? Without a full year of data it will be impossible for the authors to eliminate seasonal trends which are very strong in these similar analyses. It seems like many more months of data would be available.

See: <https://journals.plos.org/plosmedicine/article?id=10.1371/journal.pmed.1002508>

Response:

We appreciate the reviewer’s concern about the potential for seasonality of health outcomes. Although our primary hypothesis relates to the presence (or not) of an immediate interruption of services at the time of each lockdown stage implementation, we agree that there is a possibility that external shocks (e.g. budget shortfalls, school breaks) could have similar effects at the same time each year.

To respond to this comment, we have taken two additional steps to consider seasonality as a potential confounder. First, we added a lowess plots (Supplemental Figure 2) to depict changes in clinic visitation during the observation periods in 2019 and 2020, and inspect for a possibility presence of seasonality in clinic visitation. These plots show roughly consistent clinic visitation before versus after the lockdown date in 2019, while reflecting the drop in child health visits after lockdown since in 2020.

To test for a seasonality effect, we have additionally fit a differences in difference model that includes a covariate for year (2019 versus 2020), a covariate indicating pre- versus post-lockdown period, and an interaction between the two to determine if pre- to post-lockdown changes differ between years. In

these difference in difference analyses, we found similar estimates for results for the interaction term between year and pre-post period as we found in our primary analyses, suggesting that any differences in 2020 were not also observed in prior years. These results have been added to the updated Table 3.

	Mean daily visits per clinic at time of lockdown period	Stepwise change in clinic visits/day at start of level 5 lockdown	P-value
All Visits			
Difference-in-Differences ^a	96.3 (63.6, 129.0)	3.4 (-5.5, 12.4)	0.45
Childcare Visits			
Difference-in-Differences ^a	11.8 (8.0, 15.7)	-4.0 (-5.5, -2.5)	<0.001
HIV Visits			
Difference-in-Differences ^a	43.6 (25.1, 62.1)	4.8 (-0.5, 10.1)	0.08

^aDifference-in-differences estimates are estimated as the mean of the level 5 lockdown period minus the mean of the pre-lockdown period, comparing 2020 to 2019. Estimates are based on a period-by-year interaction term fit via linear mixed models.

Reviewer Comment

Why did the authors use a linear regression model to examine these count data? One would expect a Poisson or Negative Binomial model that would allow the variance to be proportional to the mean in some way. In addition, this would allow the examination of rates per population covered. Linear models do not have an intuitive residual variance estimation, can get negative predictions or counts, and do not account for difference in underlying population distributions across your 11 clinics. Please advise.

Response:

We believe that the interpretability of the linear mixed effects models would be more useful to practitioners than a rate model which is prone to misinterpretation. To assess assumptions related to the linear model selected in our primary model, we have added plots of residuals which support the normality assumption of the mean (Supplemental Figure 1). The model is specified within-clinic meaning we have an average of 100 observations per unit of observation (clinic), which further supports assumptions about the distribution of the mean.

The linear mixed models we implement also accounts for variability in several ways through the inclusion of different variance terms. The model errors account for within-subject variability about the individual mean with a parameter that is not dependent upon the mean. This detachment of the variance estimation from the mean is what is sought by the use of the overdispersion techniques the reviewer suggests whereas a Quassipoisson model induces a variance proportional to the mean and a negative-binomial model induces a variance that is a quadratic function of the mean. The random effects we implement account for betweensubject variance. The variance of the random intercept accounts for the clinic-level variation at baseline while the random slopes allow for the between-clinic

variation to vary across time. Please see our response below for more information about model specifications. In summary, given the large number of observations per center, our residual diagnostics, and our sensitivity analysis, we feel comfortable with the Gaussian assumption in light of the model producing effects that are easier to interpret.

Nevertheless, to respond to the reviewer's understandable concerns, we included a sensitivity analysis using generalized estimating equations in place of a mixed effects model to ensure that our modelling assumptions did not skew the results, and have added Poisson regression models both using mixed and GEE specifications which demonstrate similar estimates and differences after lockdown in all models (Table 3).

All Visits	Mean daily visits per clinic at time of lockdown period	Stepwise change in clinic visits/day at start of level 5 lockdown	P-value
Primary model	90.3 (67.1, 113.5)	-6.7 (-16.4, 3.0)	0.18
Poisson ^a mixed effects model	92.0 (59.3, 124.7)	-6.9 (-11.0, -2.8)	0.001
Linear GEE (exchangeable correlation matrix)	89.2 (67.0, 111.4)	-6.4 (-16.8, 4.08)	0.23
Poisson GEE ^a (exchangeable correlation matrix)	89.2 (84.7, 93.6)	-6.6 (-8.7, -4.5)	<0.001
Linear GEE (autoregressive correlation matrix)	90.2 (73.2, 107.2)	-5.4 (-27.4, 16.6)	0.63
Poisson GEE ^a (autoregressive correlation matrix)	88.4 (85.0, 91.9)	-5.0 (-9.4, -0.6)	0.03
Difference-in-Differences ^b	96.3 (63.6, 129.0)	3.4 (-5.5, 12.4)	0.45
Childcare Visits			
Primary model	11.9 (8.6, 15.1)	-7.1 (-8.9, -5.3)	<0.001
Poisson ^a mixed effects model	12.3 (7.1, 17.5)	-7.7 (-11.1, -4.4)	<0.001
Linear GEE (exchangeable correlation matrix)	11.8 (8.6, 15.0)	-7.1 (-9.0, -5.2)	<0.001
Poisson GEE ^a (exchangeable correlation matrix)	12.0 (10.5, 13.5)	-7.5 (-8.4, -6.6)	<0.001
Linear GEE (autoregressive correlation matrix)	11.9 (9.5, 14.4)	-6.4 (-9.9, -2.9)	<0.001
Poisson GEE ^a (autoregressive correlation matrix)	11.9 (10.7, 13.1)	-6.7 (-8.1, -5.4)	<0.001

Difference-in-Differences ^b	11.8 (8.0, 15.7)	-4.0 (-5.5, -2.5)	<0.001
HIV Visits			
Primary model	37.5 (24.4, 50.7)	8.0 (2.3, 13.7)	0.01
Poisson ^a mixed effects model	39.2 (22.7, 55.8)	9.0 (4.5, 13.5)	<0.001
Linear GEE (exchangeable correlation matrix)	37.7 (25.3, 50.1)	8.1 (2.2, 14.0)	0.007
Poisson GEE ^a (exchangeable correlation matrix)	37.7 (34.8, 40.6)	8.7 (7.2, 10.3)	<0.001
Linear GEE (autoregressive correlation matrix)	38.9 (29.3, 48.5)	6.1 (-6.2, 18.5)	0.33
Poisson GEE ^a (autoregressive correlation matrix)	37.9 (35.7, 40.1)	5.7 (2.6, 8.8)	0.002
Difference-in-Differences ^b	43.6 (25.1, 62.1)	4.8 (-0.5, 10.1)	0.08

Reviewer Comment

3. I do not support excluding weekends just because “most visits do not occur on weekends”. Why not just include them?

Response:

The reason for not including weekends was not clearly specified in our manuscript. Our primary outcome of interest, chronic medical care visits for adults and well-care for children, do not occur on weekends. We have updated the manuscript to reflect this:

We excluded weekends because the study clinics do not provide non-urgent ambulatory care services on weekends.

Reviewer Comment

4. Including fixed effects by day do not seem the most important to consider. What would be most important would be to eliminate seasonality at the monthly level, not weekly variations. With only 60 days of data pre-COVID, the authors are fully unable to account for seasonal variations which may be fully responsible for the changes they are seeing. This is a major limitation of these analyses.

Response:

We thank the reviewer for this suggestion. As described above, we have made two additions to the manuscript to better consider the possibility of seasonality – 1) addition of lowess plots to compare trends in clinical visitation in 2020 with 2019 and fit an additional model including year in the model to assess if changes in access differed over the period of interest between 2019 and 2020

Reviewer Comment

In the abstract as well as line 84. This is not a regression discontinuity design. In regression discontinuity, data would be assigned to intervention or control based on a cut-off score, and then analyzed in a way that compares data points just above and below the cut-off. THIS design depends on the exchangeability of the data close to the cutoff point to draw inference, where the idea in the interrupted time series design is to draw inference from all available data. So this study needs to be characterized as an uncontrolled interrupted time series analysis, not a regression discontinuity. Furthermore, this citation that is used for justification doesn't support the claim. In lines 84-85, it is the time-series design itself that allow you to investigate the immediate level change as well as the trend change.

Response:

In response to the reviewer's suggestion, we have changed the wording from a regression discontinuity to interrupted time series in the title and the text.

Reviewer Comment

Lines 102-108. Testing for autocorrelation by adding random slopes for time does not make sense. This allows for each day of the week to have a different time trend, but doesn't account for residual autocorrelation in the error term. Autocorrelation and differential trends by clinic are separate statistical and theoretical issues. I am confused. A suggestion to examine autocorrelation would be to specify an AR1 error term in the model. The authors could also add robustness to the discussion of autocorrelation by examining a PACF plot of the final model, which would reveal the presence of any additional residual autocorrelation that isn't being handled by the model.

Response:

We were incorrect in our phrasing in the use of random slopes as means of addressing autocorrelation and have removed that phrase. The inclusion of a random slope induces a time-varying covariance structure which achieves similar goals to the an AR1 without enforcing the exponential decay of the AR1 model. Thus, it does allow for a covariance structure in which between-measurement covariances are a function of time. The induced correlation of an AR1 model is $Corr(Y_{ij}, Y_{i,j+k}) = \rho^k$ where k is the distance between the measurements occurring at times j and $j + k$. The induced correlation of a linear mixed effects model with a random intercept and slope, as we employed, is more complicated but also more flexible as it allows for the correlation to be a function of time. Let t_{ij} and t_{ik} denote the times that measurements j and k occur for subject i . Further, let Y_{ij}

denote subject i 's measurement at time j . A simple linear mixed effect model with random slope and intercept then has the form

$$Y_{ij} = \beta_0 + \beta_1 t_{ij} + b_{1i} + b_{2i} t_{ij} + \epsilon_{ij}$$

where, $\epsilon_{ij} \sim N(0, \sigma^2)$, $b_{1i} \sim N(0, \sigma_{b_1}^2)$, $b_{2i} \sim N(0, \sigma_{b_2}^2)$, and $Cov(b_{1i}, b_{2i}) = \sigma_{b_1, b_2}$.

The induced marginal variance is then

$$Var(Y_{ij}) = \sigma_{b_1}^2 + 2t_{ij}\sigma_{b_1, b_2} + t_{ij}^2\sigma_{b_2}^2 + \sigma^2.$$

Similarly, the induced marginal covariance is

$$Cov(Y_{ij}, Y_{ik}) = \sigma_{b_1}^2 + (t_{ij} + t_{ik})\sigma_{b_1, b_2} + t_{ij}t_{ik}\sigma_{b_2}^2.$$

The correlation, having the form

$$\frac{\sigma_{b_1}^2 + (t_{ij} + t_{ik})\sigma_{b_1, b_2} + t_{ij}t_{ik}\sigma_{b_2}^2}{\sqrt{\sigma_{b_1}^2 + 2t_{ij}\sigma_{b_1, b_2} + t_{ij}^2\sigma_{b_2}^2 + \sigma^2} \sqrt{\sigma_{b_1}^2 + 2t_{ik}\sigma_{b_1, b_2} + t_{ik}^2\sigma_{b_2}^2 + \sigma^2}}$$

is a nonlinear function of both t_{ij} and t_{ik} as well as a nonlinear function of the within-subject variability and between-subject variabilities. Thus, the induced covariance structure we implement is indeed time-dependent and accounts for within-sample correlation.

Reviewer Comment

Also, the authors fit GEE and poisson GEE models for sensitivity analysis. What type of working correlation matrix was used in the GEE models? Exchangeable? AR1? Fitting a GEE model is quite a different approach and results in a marginal model, whereas the GLMMs are conditional models. A more rational sensitivity analysis would be to fit Poisson or Negative Binomial GLMMs and compare them to the Linear GLMM model. Please advise.

Response:

To response to this potential concern about mis-specifying the correlation matrix, in this version of the manuscript, we have fit four GEE models – a linear model and Poisson model, using both exchangeable and autoregressive correlation matrices. These show similar estimates for the difference in clinic visits/day for our primary outcomes in all models (Table 3):

	Mean daily visits per clinic at time of lockdown period	Stepwise change in clinic visits/day at start of level 5 lockdown	P-value
All Visits			
Primary model	90.3 (67.1, 113.5)	-6.7 (-16.4, 3.0)	0.18
Poisson ^a mixed effects model	92.0 (59.3, 124.7)	-6.9 (-11.0, -2.8)	0.001
Linear GEE (exchangeable correlation matrix)	89.2 (67.0, 111.4)	-6.4 (-16.8, 4.08)	0.23
Poisson GEE ^a (exchangeable correlation matrix)	89.2 (84.7, 93.6)	-6.6 (-8.7, -4.5)	<0.001
Linear GEE (autoregressive correlation matrix)	90.2 (73.2, 107.2)	-5.4 (-27.4, 16.6)	0.63
Poisson GEE ^a (autoregressive correlation matrix)	88.4 (85.0, 91.9)	-5.0 (-9.4, -0.6)	0.03
Difference-in-Differences ^b	96.3 (63.6, 129.0)	3.4 (-5.5, 12.4)	0.45
Childcare Visits			
Primary model	11.9 (8.6, 15.1)	-7.1 (-8.9, -5.3)	<0.001
Poisson ^a mixed effects model	12.3 (7.1, 17.5)	-7.7 (-11.1, -4.4)	<0.001
Linear GEE (exchangeable correlation matrix)	11.8 (8.6, 15.0)	-7.1 (-9.0, -5.2)	<0.001
Poisson GEE ^a (exchangeable correlation matrix)	12.0 (10.5, 13.5)	-7.5 (-8.4, -6.6)	<0.001
Linear GEE (autoregressive correlation matrix)	11.9 (9.5, 14.4)	-6.4 (-9.9, -2.9)	<0.001
Poisson GEE ^a (autoregressive correlation matrix)	11.9 (10.7, 13.1)	-6.7 (-8.1, -5.4)	<0.001
Difference-in-Differences ^b	11.8 (8.0, 15.7)	-4.0 (-5.5, -2.5)	<0.001
HIV Visits			
Primary model	37.5 (24.4, 50.7)	8.0 (2.3, 13.7)	0.01
Poisson ^a mixed effects model	39.2 (22.7, 55.8)	9.0 (4.5, 13.5)	<0.001

Linear GEE (exchangeable correlation matrix)	37.7 (25.3, 50.1)	8.1 (2.2, 14.0)	0.007
Poisson GEE ^a (exchangeable correlation matrix)	37.7 (34.8, 40.6)	8.7 (7.2, 10.3)	<0.001
Linear GEE (autoregressive correlation matrix)	38.9 (29.3, 48.5)	6.1 (-6.2, 18.5)	0.33
Poisson GEE ^a (autoregressive correlation matrix)	37.9 (35.7, 40.1)	5.7 (2.6, 8.8)	0.002
Difference-in-Differences ^b	43.6 (25.1, 62.1)	4.8 (-0.5, 10.1)	0.08

Reviewer Comment

Lines 102-8. Sensitivity analysis that aims to allow for non-linear functional form by adding a quadratic term to the model: This would allow curvature in the data, but I'm not convinced this is the most appropriate way to assess for non-linearity, especially given the scatterplots don't display a non-linear relationship. Furthermore, if the relationship pre-COVID-19 is a non-linear relationship, the ability to forecast this as a counterfactual for "what would have happened" in the absence of COVID-19 is compromised.

Response:

We agree with the reviewer that, based on the data we have low suspicion for a non-linear effect, and are less interested in trends over time than the level change after the lockdown. We have thus removed the quadratic term sensitivity analyses from the manuscript.

Reviewer Comment

General. Please provide your full statistical model to make clear how you parameterized the time-series elements of your effect (immediate level change and slope change).

Response: Our full statistical model can be displayed here:

Let y_{ij} be the number of visits to the i th clinic on the j th day. The primary model is then described by:

$$y_{ij} = \beta_0 + \beta_1 \text{Level5}_{ij} + \beta_2 \text{Level4}_{ij} + \beta_3 \text{Level3}_{ij} + \beta_4 \text{Time}_{ij} + \beta_5 \text{Level5}_{ij} \times \text{Time}_{ij} + \beta_6 \text{Level4}_{ij} \times \text{Time}_{ij} + \beta_7 \text{Level3}_{ij} \times \text{Time}_{ij} + \beta_8 \text{Tues}_{ij} + \beta_9 \text{Wed}_{ij} + \beta_{10} \text{Thurs}_{ij} + \beta_{11} \text{Fri}_{ij} + b_{i0} + b_{i1} \text{Time}_{ij} + \varepsilon_{ij},$$

where Level5_{ij} , Level4_{ij} , and Level3_{ij} are indicator variables for the day being during Level 5, Level 4, or Level 3, respectively. Time_{ij} is a time variable centered at the start of the lockdown. Tues_{ij} , Wed_{ij} , Thurs_{ij} , and Fri_{ij} are indicator variables for their respective days of the weeks. We assume the random intercept, b_{i0} , random slope of time, b_{i1} , and model error, ε_{ij} , are normally distributed: $b_{i0} \sim N(0, \sigma_{b0}^2)$, $b_{i1} \sim N(0, \sigma_{b1}^2)$, and $\varepsilon_{ij} \sim N(0, \sigma_{\varepsilon}^2)$. Further, we assume the following variance-covariance matrix on the vector of random effects

$$\text{Var} \left(\begin{bmatrix} b_{i0} \\ b_{i1} \end{bmatrix} \right) = \begin{bmatrix} \sigma_{b0}^2 & \sigma_{b0,b1} \\ \sigma_{b0,b1} & \sigma_{b1}^2 \end{bmatrix},$$

where $\sigma_{b0,b1}$ is the covariance, i.e. $\text{Cov}(b_{i0}, b_{i1})$.

Reviewer Comment

General. Were all of these analyses post-hoc or pre-specified? Please make absolutely clear what was prespecified and what was decided upon after graphing the data, looking at the data, fitting alternative models, etc. A roadmap of exactly the steps the authors took to the analysis would be helpful.

Response:

For the initial submission, we followed a pre-specified analysis plan. In response to editorial and reviewer feedback, and with rapidly changing to the circumstances since the first submission (i.e. change from level 5 to level 4 to level 3 lockdown) we added new analyses after the initial analysis plan was formulated. We have clarified this in the manuscript:

Three study investigators designed a statistical plan prior to all analyses (MJS, JDK, MDM). The initial analysis plan included fitting mixed effects models with random effects by clinic and inspecting trends in clinic visitation changes from the pre- to level 5 lockdown period for the overall cohort, and by visit subtype. We also initially included plans for sensitivity analyses, including fitting of generalized estimating equations and additions of random slopes to our models as robustness checks. In response to reviewer requests and with updates to the lockdown characteristics from levels 5 to 4 to 3 during the review process, we conducted a number of post-hoc analyses, including construction of loess plots and fitting additional models to assess for seasonal trends in visitation by year, and fitting mixed effects Poisson models. All statistical analyses were conducted using Stata and R.

Reviewer Comment

Table 2. Please modify the way you describe your immediate level change. The way it is currently described “change in mean clinic visits per day” makes it sound like a change in slope rather than an immediate change in level. It is just the mean change in clinic visits immediately after the lockdown (change in intercept). There is no time component.

Response:

We have updated the wording of the header categories in Table 2 to better specify that we are measuring the stepwise change at the time of transition from pre-lockdown to level 5, and from level 5 to level 4 and level 4 to level 3 transitions.

Reviewer Comment

Supplementary Table 3. It appears as though the Poisson GEE model is highly significant, where the primary model, random slope model, and Linear GEE models are not significant. Thus, it seems possible that many of the changes you declare are “not statistically significant” may in fact have been significant had you used a count-based model rather than a linear model. Please advise. One issue with Poisson models is overdispersion. I would highly suggest using either a quasi-Poisson model that allows for overdispersion, or a negative binomial model that builds the flexibility for overdispersion in the model.

Response:

We have added a Table 3 (displayed above) to more clearly show the robustness of our results with each of the difference sensitivity analyses. Although the GEE model is statistically significant, the predicted numeric change in clinic visits is nearly identical to the linear model, and we believe accurately reflects a nonsignificant change from prior to lockdown. This is further evidenced by the plots which also do not show a change in visitation over time.

Reviewer Comment

Figures on Page 32. I assume the line represents the overall mean of the model and the shaded gray represents Cis? Please improve these models by doing the following:

1. Add the true predicted value from your model which would include the daily fixed effects (this should vary by day).
2. Keep the overall linear mean (so the daily fixed effect should vary around the overall mean)
3. Add in the raw mean value by day across all clinics, rather than a forest plot of all observations.

These three changes will allow us to see whether your Cis are appropriate and whether your model is adequately fitting to the raw mean over time. The current figures do not allow this (this is essentially the best “validation” one can do in a time-series analysis).

Response:

As suggested by the reviewer we have updated our figures to include both our model fit line and the true mean of the visits on each day:

As described above, we have also added lowess plots (Supplemental Figure 2) to depict changes without restricting to our model assumptions.

Reviewer Comment

Discussion line 192. “expected 20% increase in clinic visits for HIV”. Please replace with “estimated 20% increase in clinic visits for HIV”? Or observed? It is not expected, which would refer to a value in the future or an out of sample prediction.

Response:

We have removed that phrase from the manuscript as well as references to expected estimates.

Reviewer Comment

Discussion generally. The authors should add a discussion as to why they believe clinic visits decreased so dramatically for ages ≤ 5 but remained constant for other ages?

Response:

We agree this is a critical question. Our dataset is not positioned to answer this question, although we hope to delve into it further in future qualitative and household surveys in the catchment area. We have added further text to the discussion to raise possible hypotheses:

Although these data do not suggest the cause of reduced visitation in children, multiple possible factors might be considered. We hypothesize that limited options for childcare for families with multiple children might prevent caregivers from being able to bring individual children to clinic. Moreover, in contrast to, for example, HIV wellcare visits, well child visits rarely involve medication refills so might be prioritized lower for families. Whatever the cause, our findings are in keeping with data from elsewhere. In Hangzhou, China, pediatric healthcare visits dropped by nearly 75% during the peak of the epidemic and lockdown periods.³⁸ In the U.S. vaccination rates in children substantially declined after a national emergency was declared in response to the COVID-19 epidemic.³⁹

Reviewer: 2

Reviewer Name: Sunday Olawale Onagbiye

Institution and Country: University of the Western Cape, South Africa

Please state any competing interests or state 'None declared': None

Reviewer Comment

In line 27: this statement seems standalone and seems not related to the outcome investigated and should be removed for clarity. on the other hand since the the study is not focusing on Ebola.

Response:

We agree with the reviewer that this statement was not well placed. We have attempted to add context to the sentence to demonstrate that we might learn from prior infectious disease epidemics that also reduced access to primary care. We would be happy to remove the sentence altogether if the reviewer continues to find it out of place:

There is also historical precedent from other recent communicable disease outbreaks. Primary healthcare access was significantly impacted during prior infectious disease epidemics, such as Ebola virus disease, resulting in increases in morbidity and mortality.^{30 31} Yet, whether and the extent to which similar effects will be seen during the COVID-19 epidemic is not known.

Reviewer Comment

In line 33: The author seems to mostly attached social distancing to level 5 lock-down alone. it is important for the author to understand that social distancing is still existing at level 3 lock-down, although a bit of relaxing the lockdown measures. This statement needs to be rephrased for clarity for those who may the reader of already a good manuscript.

Response:

We agree. The manuscript was written just after the end of the level 5 lockdown. Since the prior submission, we have expanded the analysis to include more recent changes to level 4 and level 3 lockdowns and have reworded this section to better describe the range of regulations:

On 27th March, 2020, South Africa instituted a nationwide shelter-in-place order, termed in South Africa as a Level 5 lockdown The level 5 order included closure of schools and all non-essential business, restrictions on public transport, and restrictions on movement. Restrictions on movement during the level 5 lockdown specifically required that individuals remain in their place of residence, with the exceptions of “performing an essential service, obtaining an essential good or service, collecting a social grant, pension, or seeking emergency, life-saving or chronic medical attention.” Over the following months, the restrictions gradually eased from level 5 down to 4 at the end of April and level 3 at the end of May, which corresponded with lifting restrictions on intra-province movement, preinitiation of public transportation, and allowed for reopening of schools and many business. ^{32 33} Because the healthcare sector was deemed an essential service throughout the entire lockdown period, no restrictions were placed on access to or delivery of healthcare services.

Reviewer Comment

In line 36: This statement seems not so important here. This is what was already known as essential services.

Response:

We agree and, as described in the prior comment, added information to better contextualize each stage of the lockdown in South Africa.

Reviewer Comment

In line 38: Before the start of the statement "We sought....before the next statement that leads to why the study is conducted, I suggest the author should add 1-2 paragraph on access to primary health care in south Africa before lockdown, challenges during the lock-down, and this can then be followed with what was done in the study. This could later inform the re-organisation of the introduction section.

Response:

We thank the reviewer for this suggestion and have significantly added to this section, with additional updates from the literature since our prior submission:

Of particular concern is how social fear and reduced access to basic public health services might impact morbidity and mortality for non-COVID health conditions. Modeling studies have suggested that even modest reductions in child healthcare access could result in 100,000s of additional deaths in low and middle-income countries.²³ Similar concerns have been raised by the Academy of Science of South Africa and others about provision of chronic disease care among adults.^{24 25} UNAIDS has warned that nonpharmaceutical interventions could challenge manufacturing and supply chains of HIV therapeutics,²⁶ and modeling estimates suggest that such disruptions could cause as many if not more HIV-related deaths than COVID-19-related deaths.²⁷ Although empiric data on health outcomes remain sparse, there have been significant reductions in tuberculosis testing in South Africa during the early phases of the lockdown,²⁸ indicating an interruption in critical services for the most common cause of death in the country.²⁹ There is also historical precedent from other recent communicable disease outbreaks. Primary healthcare access was significantly impacted during prior infectious disease epidemics, such as Ebola virus disease, resulting in increases in morbidity and mortality.^{30 31} Yet, whether and the extent to which similar effects will be seen during the COVID-19 epidemic is not known.

Reviewer Comment

In line 51: the "p" should be capitalized

Response:

Thank you – we have done so.

Reviewer Comment

In line 66: this statement should be rephrased

Response: Thank you – we have updated this sentence to read:

We link data between the HDSS and the clinic medical record system electronically using a unifying identification code for each resident of the catchment area.

Reviewer Comment

In line 123: contributed "to" the study design and collection measures (delete input on)

Response:

We have deleted that word:

This protocol was reviewed and approved by the AHRI Community Advisory Board, who contributed to the study design and selection of collection measures

Reviewer Comment

in line 142: "years old" should come after (n=3173).

Response:

We have updated this sentence:

Approximately 9% of visits were made by individuals less than 1 year old (n=4,186), 1-5 year old (n=3,944), and 6-19 years old (n=4,460), respectively...

Reviewer Comment

in line 143: the writing of result section should follow the same pattern for clarity. it is good to write the percentage accounted for follow by number of participants e.g. 47% (n=17,226). This should be considered for others. Same is applicable for the regressions results that follows.

Response:

We agree and have included percentages and n for all proportions and the 95%CI around all mean estimates.

Reviewer Comment

in line 231: this statement should be referenced or the source should be provided.

Response:

We have added the following reference to support this sentence:

Wilkinson L, Grimsrud A. The time is now: expedited HIV differentiated service delivery during the COVID-19 pandemic. Journal of the International AIDS Society 2020;23(5) doi: 10.1002/jia2.25503

Reviewer Comment

in line 234: this statement is not clear and should be given attention

Response:

We have updated the sentence to add clarity:

The lockdown was instituted rapidly in South Africa, before substantial decentralized care systems could be put in place.

Reviewer Comment

in line 248: instead of the word "will", I suggest the author use the word "may"

Response:

We have made the suggested change:

Mitigating longer-term consequences may require governments and development partners to increase access to employment and other social support services during the epidemic

Reviewer Comment

in line 252: Not sure if this could be a strong motivation for strength of the study. the author should think about this.

Response:

We feel that our study is strengthened by the presence of independent data collectors who might be less susceptible to human resource interruptions at the time of the epidemic compared to other programmatic or public health management data collection systems. We would be happy to discuss this further with the reviewer if there are concerns about this point.

Reviewer Comment

in line 271: "here" should be written as "in this study"

Response:

We have made the suggested edit: Future work should establish if these trends are maintained, and particularly monitor access to childcare and immunizations as a result of the trends reported in this study.

Reviewer Comment

in line 272: the final part as highlighted here does not seem to be a suggestion. I am also concerned about the word "will" instead of using words such as may; might; or could. This should be rephrased.

Response:

We agree that this could have been better stated. We have edited that sentence to read:

Finally, in rural South Africa and similar areas, efforts to balance ongoing provision of essential preventative and chronic healthcare services might be needed to ensure healthcare access remains intact while preventing nosocomial spread of COVID-19 among high-risk populations.

Reviewer Comment

In table 1: the author should check for the correctness and accuracy of the percentages, especially when it is added up which must be 100%. e.g. in a situation where the add up could be 100.2% instead of 100%.

Response:

These percentages are not necessarily going to add up to 100%. Individuals are able to give more than one reason for presenting to care. We have better specified this in the table:

*Visit types are not mutually exclusive so column totals may exceed 100%

Reviewer Comment

Lastly, this article should be language edited by English language editor or professional in the field. Importantly, the professional in the field would be preferred in order not to lose the scientific writing and language.

Response:

Thank you. We have thoroughly reviewed and edited the document for English language and grammar.

VERSION 2 – REVIEW

REVIEWER	BRADLEY H WAGENAAR Department of Global Health, University of Washington, United States
REVIEW RETURNED	08-Sep-2020

GENERAL COMMENTS	The reviewer completed the checklist but made no further comments.
--

REVIEWER	Sunday Olawale Onagbiye University of the Western Cape South Africa
REVIEW RETURNED	24-Aug-2020

GENERAL COMMENTS	1. The author needs to check all references for completeness. I think some references are incomplete. It is important that author cited get full credit by proper citations. For example, reference number below; *12. Peak CM, Kahn R, Grad YH, et al. 2020 doi: 10.1101/2020.03.05.20031088 *18. Mongey S, Pilossoph L, Weinberg A. 2020 doi: 10.3386/w27085 *19. Quaife M, van Zandvoort K, Gimma A, et al. 2020 doi: 10.1101/2020.06.06.20122689 *27. Jewell BL, Smith JA, Hallett TB. 2020 doi
--

VERSION 2 – AUTHOR RESPONSE

2. The author needs to check all references for completeness. I think some references are incomplete. It is important that author cited get full credit by proper citations. For example, reference number below;

*12. Peak CM, Kahn R, Grad YH, et al. 2020 doi: 10.1101/2020.03.05.20031088

*18. Mongey S, Pilossoph L, Weinberg A. 2020 doi: 10.3386/w27085

*19. Quaife M, van Zandvoort K, Gimma A, et al. 2020 doi: 10.1101/2020.06.06.20122689

*27. Jewell BL, Smith JA, Hallett TB. 2020 doi

We have reviewed and updated all of our references to include full details of the publications as suggested